# A Practical Approach to Multimodality Imaging in Hypertrophic Cardiomyopathy

**DOI:** 10.3390/jcm14082606

**Published:** 2025-04-10

**Authors:** Ankur K. Dalsania, Christine M. Park, Sanjana Nagraj, Daniel Lorenzatti, Annalisa Filtz, Adaya Weissler-Snir, Mario J. Garcia, Leandro Slipczuk, Aldo L. Schenone

**Affiliations:** Montefiore Einstein Center for Heart & Vascular Care, Montefiore Medical Center, Albert Einstein College of Medicine, Bronx, NY 10461, USA; adalsania@montefiore.org (A.K.D.); chrpark@montefiore.org (C.M.P.); snagraj@montefiore.org (S.N.); dlorenzatt@montefiore.org (D.L.); afiltz@montefiore.org (A.F.); aweisslers@montefiore.org (A.W.-S.); mariogar@montefiore.org (M.J.G.); lslipczukb@montefiore.org (L.S.)

**Keywords:** hypertrophic cardiomyopathy, multimodality imaging, imaging targets

## Abstract

Hypertrophic cardiomyopathy remains underdiagnosed despite a growing number of effective treatment interventions that can improve care. Multimodality imaging has become integral to diagnosing and managing hypertrophic cardiomyopathy, providing a comprehensive assessment of the disease. In particular, it enhances the diagnostic accuracy and deepens the understanding of the mechanisms underlying patient symptoms, enabling targeted therapeutic approaches. Additionally, multimodality imaging allows for better risk stratification, assessment of therapy response, and guidance of interventions to deliver personalized medicine. The practical tools outlined in this review can help providers integrate multimodality imaging strategies to provide better care and improve the patient experience.

## 1. Introduction

Hypertrophic cardiomyopathy (HCM) is a genetic disease of the myocardium characterized by left ventricular hypertrophy (LVH) without an alternative cause capable of explaining the magnitude of hypertrophy [1,2]. HCM has a wide range of disease penetrance, clinical expressions, and morphological patterns, contributing to its challenging diagnosis [3]. Even though the prevalence is estimated to range from 1:500 to 1:200, HCM remains underdiagnosed, with only about 10% carrying the diagnosis [4,5]. Establishing the diagnosis of HCM is essential as there are effective treatment interventions that improve morbidity and mortality [3,6,7]. Even among those diagnosed, many continue to experience symptoms and major cardiovascular events due to limited discrimination of underlying disease mechanisms that prevent the initiation of targeted treatments.

Multimodality imaging plays a central role in the contemporary management of HCM. By integrating various imaging techniques to manage patients with or at risk for HCM, clinicians can comprehensively understand the disease process, thereby optimizing diagnostic accuracy, refining risk stratification, and guiding personalized treatment strategies. This review provides a practical guide to utilizing multimodality imaging in HCM.

## 2. Diagnostic Challenges and Imaging Targets in HCM

No single clinical or imaging finding in isolation can definitively diagnose HCM. While increased maximal left ventricular wall thickness is a primary diagnostic criterion, its presence, even with dynamic left ventricular outflow tract obstruction (LVOTO), is not independently diagnostic [2]. Over-reliance on these findings in isolation can lead to overdiagnosis, as these may be present in HCM mimickers, including hypertensive heart disease, infiltrative cardiomyopathies (e.g., cardiac amyloidosis), and metabolic storage disorders [8]. Conversely, underdiagnosis remains a significant issue in HCM care. This often stems from the failure to identify localized LVH or misinterpreting asymmetric hypertrophy as concentric hypertrophy, a pattern usually attributed to other conditions.

Evaluation of suspected HCM necessitates a comprehensive approach that integrates imaging findings with clinical data. From an imaging perspective, accurate diagnosis and characterization of HCM requires moving beyond simple wall thickness measurement and assessment of LVOTO to a nuanced understanding of the diverse pathophysiological processes at play. Non-invasive imaging can directly or indirectly evaluate these processes and provide important diagnostic, therapeutic, and prognostic implications.

### 2.1. Left Ventricular Hypertrophy and Morphology

LVH is the primary diagnostic feature of HCM. In adults, a diagnosis is suggested by a maximal end-diastolic wall thickness ≥15 mm anywhere in the left ventricle or 13–14 mm with a family history of HCM or a known pathogenic/likely pathogenic sarcomere mutation, in the absence of other causes of LVH [1,2]. For children, a left ventricle (LV) wall thickness body surface area-adjusted z-score ≥ 2 standard deviations above the mean is used as a diagnostic criterion (Table 1) [9]. It is essential to recognize that reliance on fixed wall thickness thresholds may lead to underdiagnosis, especially in cases of subthreshold hypertrophy or apical HCM variants. Emerging research advocates using demographic-adjusted wall thickness cut-offs to enhance diagnostic accuracy and case finding [10]. Moreover, an indexed apical wall thickness exceeding 5.6 mm/m^2^, accompanied by the absence of normal apical tapering, has been proposed as a more sensitive marker for identifying apical HCM [10,11].

The pattern of hypertrophy is often asymmetric, typically involving less than half of the LV segments and occasionally confined to one or two segments [12]. The most common pattern is focal asymmetric hypertrophy of the basal anterior septum. Distinct LVH phenotypes have important clinical implications. LVOTO most likely accompanies the sigmoid phenotype, whereas the reverse curvature type is most likely to carry sarcomere mutations and exhibit late gadolinium enhancement (LGE) [13]. The apical variant is regarded as a more benign phenotype in terms of sudden cardiac death (SCD). Importantly, increased LV mass is not a diagnostic prerequisite, as focal hypertrophy can occur with a normal overall LV mass. Beyond diagnosis, LV wall thickness has prognostic value with a parabolic relationship between the degree of LV thickening and symptom severity. Furthermore, extreme LV wall thickness (>28–30 mm) is a strong predictor of SCD [2].

### 2.2. Systolic Function and Myocardial Deformation

The left ventricular ejection fraction (LVEF) is usually normal to hyperdynamic in patients with HCM, which is explained by an exaggerated contractile force generation in response to increased actin–myosin cross-bridges. Nonetheless, these patients demonstrate paradoxical contractile dysfunction, as demonstrated by an impaired global longitudinal strain (GLS), a more sensitive marker of contractile dysfunction than LVEF [9]. Lower (less negative) GLS values are of prognostic value and predict the occurrence of HCM-related adverse outcomes. It is unclear whether abnormal GLS reflects the underlying hypertrophy, myocardial disarray, or myocardial fibrosis. Finally, a reduced LVEF < 50% denotes progression to end-stage disease with poor prognosis and increased risk of SCD [14].

### 2.3. Diastolic Dysfunction

Diastolic dysfunction is present in approximately 80% of patients with HCM, irrespective of their morphological phenotype. This is characterized by elevated filling pressures and delayed myocardial relaxation due to myocardial hypertrophy, fibrosis, microvascular ischemia, and altered energetics [15]. In some cases, increased stiffness and severe hypertrophy significantly reduce the ventricular cavity size and stroke volume, potentially leading to restrictive physiology [16]. Notably, diastolic dysfunction can precede the development of overt LVH and independently contributes to reduced exercise capacity and adverse prognosis.

### 2.4. Apical Aneurysm

Left ventricular apical aneurysms, which occur in 2–5% of cases, particularly in apical and mid-cavitary phenotypes, are crucial imaging targets in HCM risk stratification [17]. These aneurysms result from repetitive myocardial ischemia, leading to transmural scarring and subsequent aneurysmal degeneration. Current guidelines consider apical aneurysms of any size a significant risk factor when evaluating for implantable cardioverter defibrillator (ICD) therapy for primary prevention [2]. However, recent evidence suggests that a larger aneurysm size (≥2 cm) is associated with a poorer prognosis and a higher risk of SCD [18].

### 2.5. Left Ventricular Outflow Tract Obstruction

Approximately 70% of patients with HCM exhibit LVOTO, either at rest (35%) or with provocation (35%) [1,2]. The remaining 30% have non-obstructive HCM. While a peak LVOT gradient ≥ 30 mmHg defines obstruction, symptoms often manifest with gradients ≥ 50 mmHg. Notably, the presence of LVOTO is linked to an increased risk of SCD and heart failure progression in patients with HCM [19].

The primary mechanism of LVOTO in HCM is systolic anterior motion (SAM) of the mitral valve, a dynamic obstruction influenced by ventricular loading conditions and contractility. SAM results from a complex interplay of forces within the LVOT, often involving septal hypertrophy, LVOT narrowing, hyperdynamic LV, and mitral valve abnormalities [17]. Although common in HCM, SAM can occur in other conditions such as hypertensive heart disease and after mitral valve repair [20]. Mid-cavitary obstruction, alone or combined with LVOTO, can also occur due to hypertrophied or anomalous papillary muscles and a hyperdynamic LV with systolic cavity obliteration [21].

### 2.6. Abnormalities in Mitral Valve Apparatus and Mitral Regurgitation

Structural abnormalities of the mitral valve apparatus are common in HCM and can play a significant role in developing LVOTO and mitral regurgitation (MR). In patients with HCM, mitral valve abnormalities frequently include leaflet elongation, direct insertion of anomalous papillary muscles into the anterior mitral leaflet or LVOT, and anterior or apical displacement of the papillary muscles. MR in HCM can be secondary to SAM of the mitral valve or arise from intrinsic mitral valve pathology, such as mitral valve prolapse. Significantly, factors that exacerbate LVOTO severity often concurrently worsen the severity of SAM-related MR.

### 2.7. Myocardial Ischemia and Microvascular Dysfunction

Myocardial ischemia is common in patients with HCM, even without obstructive epicardial coronary artery disease, and is virtually universal in apical HCM [22]. This is primarily due to a supply–demand mismatch. Reduced myocardial oxygen supply in HCM results from decreased intramural arteriole density (rarefaction) with medial hypertrophy, microvascular dysfunction with impaired coronary flow reserve, and elevated filling pressures that reduce coronary perfusion [2,23]. Simultaneously, myocardial oxygen demand is increased by hypertrophy and intrinsically higher sarcomere-level force generation [23]. Myocardial bridging, more prevalent in HCM, can further exacerbate ischemia [1].

### 2.8. Myocardial Fibrosis

Myocardial fibrosis carries significant prognostic implications. It exists in two forms: interstitial and replacement. Interstitial fibrosis may represent a maladaptive response to chronic energy deprivation and supply–demand mismatch, attempting to preserve the cardiac structure. However, replacement fibrosis results from repetitive myocardial injury, leading to myocyte death and irreversible scarring [24]. This type of fibrosis is found in 50–60% of HCM patients, typically in the most hypertrophied segments, with a characteristic patchy, mid-myocardial distribution [17]. The extent of replacement fibrosis directly correlates with ventricular arrhythmias and SCD risk [25].

## 3. Multimodality Imaging in the Evaluation of HCM

### 3.1. Echocardiography

Transthoracic echocardiography (TTE) is the first-line imaging modality in evaluating all patients with known or suspected HCM (Table 2 and Table 3). Transesophageal echocardiography (TEE) is indicated in cases of suboptimal TTE imaging to elucidate LVOTO etiology or MR mechanism if uncertain, or intraoperatively during septal reduction therapies.

LVH assessment requires describing the pattern and distribution of hypertrophy and measuring maximal wall thickness. TTE can miss localized hypertrophy, especially in the basal anteroseptal, apical, and lateral areas, and ultrasound-enhancing agents (UEAs) can improve the detection of these patterns. Conversely, TTE can overestimate wall thickness due to foreshortening, tangential cuts, or inclusion of the RV trabeculae or moderator band (Figure 1) [1]. Integrating short- and long-axis views ensures accurate measurements.

LVEF should be evaluated at the initial diagnosis, with changes in clinical status, or during the surveillance of patients on cardiac myosin inhibitors (CMIs). Strain analysis and GLS measurements offer valuable prognostic data and should be considered in centers with experience in strain echocardiography. Strain may also aid in the discrimination of HCM from phenocopies such as cardiac amyloidosis (Figure 2). A comprehensive assessment of diastolic function should be routinely performed and include parameters such as the E/e’ ratio, left atrial volume index (LAVi), the duration difference between pulmonary vein atrial reversal (Ar) velocity and mitral inflow A-wave duration (Ar-A duration), and the peak tricuspid regurgitation (TR) jet velocity [9,26]. These parameters are applicable regardless of LVOTO; however, the E/e’ ratio and LAVi are unreliable with moderate or severe MR. Conventional E/e’ cut-off values may not directly apply to HCM; up to 25% of HCM patients with elevated E/e’ may have normal filling pressures. However, a restrictive filling pattern with elevated E/e’ ratio correlates with adverse outcomes, including heart failure hospitalizations, reduced exercise tolerance, and SCD [17].

Thorough LVOTO assessment, including provocative maneuvers, is needed to differentiate obstructive from non-obstructive phenotypes and distinguish dynamic LVOTO, which is characteristic of HCM, from fixed sub-valvular, valvular, or supravalvular aortic stenosis. Evaluation should identify and grade the SAM of the mitral valve using M-mode or 3D bi-plane imaging. The obstruction’s presence, severity, and location (i.e., LVOT, mid-cavitary, or multi-level) should be determined using color and pulse-wave Doppler (Figure 3). Factors contributing to obstruction, such as septal hypertrophy, LVOT narrowing, hyperdynamic LVEF, and mitral valve abnormalities, should be carefully assessed.

Continuous-wave (CW) Doppler is used to measure the peak instantaneous LVOT gradient, characteristically a late-peaking, “dagger-shaped” signal (Figure 3). Assessments should be performed at rest and if the resting gradient is <50 mmHg, with provocative maneuvers (Valsalva, squat-to-stand, or amyl nitrite) (Figure 3). Severe LVOTO (typically > 60 mmHg at rest) may show a “lobster-claw” spectral Doppler pattern due to a mid-systolic drop in LV ejection velocities. Differentiating LVOT signals from MR can be challenging, especially with late systolic MR. Velocities exceeding 5.5 m/s likely represent MR rather than LVOTO (Figure 3). Several techniques can aid this distinction (Table 4). In complex cases, the gradient can be indirectly estimated using the MR jet and systolic blood pressure (SBP) (LVOT gradient = SBP – 4 × [peak MR velocity]^2^ + 15–20 mmHg) but should be applied cautiously due to reliance on assumptions.

It is also essential to define the underlying mechanisms and severity of MR in HCM (SAM-related vs. primary MR) [27]. SAM-related MR typically, though not invariably, results in a posteriorly or laterally directed eccentric jet [28]. Quantifying the severity of eccentric MR jets can be challenging, often necessitating volumetric assessment of regurgitant volume and fraction.

Key features of apical aneurysms include a discrete, thin-walled (often ≤5 mm), dyskinetic, or akinetic region at the LV apex, clearly demarcated from the adjacent myocardium, frequently with a diastolic bulge [18]. Pitfalls such as foreshortening and misinterpretation of other conditions such as takotsubo cardiomyopathy must be avoided. A low threshold should be established for utilizing UEAs to improve the visualization of apical aneurysms and detect apical thrombi.

### 3.2. Stress Echocardiography

Exercise stress echocardiography is primarily indicated in symptomatic patients with HCM who exhibit a resting or provoked peak LVOT gradient < 50 mmHg on TTE to unmask latent LVOTO capable of producing symptoms (Table 2 and Table 3) [2]. Upright exercise is the preferred method as it provides the greatest physiological stimulus for provoking LVOTO and typically induces higher gradients than supine exercise [1]. Conversely, dobutamine stress echocardiography is not recommended for provoking LVOTO in HCM due to its non-physiological effects and potential for inducing dynamic gradients even in normal individuals [1,3].

Beyond its role in provoking LVOTO, exercise stress echocardiography may provide valuable insights into a patient’s functional capacity and symptom status under physiological stress. It also allows for assessing exercise-induced myocardial ischemia, changes in the severity of MR, and diastolic function parameters. While the role of diastolic parameters in exercise is still under investigation, assessing these parameters may help identify the mechanism of exercise intolerance and influence management decisions. It is reasonable to repeat exercise stress echocardiography if there is a decline in functional status or every 2–3 years in asymptomatic patients, especially if there is ambiguity regarding changes in their functional capacity [2].

### 3.3. Cardiac Magnetic Resonance Imaging

Cardiac magnetic resonance (CMR) is crucial for HCM diagnosis, risk stratification, and procedural planning, complementing echocardiography in most suspected or confirmed cases (Table 2 and Table 3). This modality more reliably identifies subtle or atypical patterns of LVH and apical aneurysms, often missed by echocardiography (Figure 4) [29,30]. It offers a more precise measurement of LV wall thickness, which is crucial for guiding ICD implantation by avoiding overestimation from tangential septal measurements or right ventricular trabeculae (Figure 1). CMR also differentiates HCM from phenocopies like cardiac amyloidosis, Fabry’s cardiomyopathy, athlete’s heart, and hypertensive heart disease, among others (Figure 5) [9,31,32].

CMR is the gold standard for evaluating LVEF, typically indicated when echocardiographic assessment is non-diagnostic or equivocal. This modality could also provide valuable insights into diastolic function by assessing LV mass, left atrial (LA) size, LV and LA strain, and parameters like peak filling rate (PFR) and time to peak filling rate (TPFR). Emerging evidence suggests that 4D flow techniques may allow the evaluation of mitral inflow patterns as part of the evaluation of diastolic function. CMR can further elucidate the factors contributing to LVOTO (Figure 6). While phase-contrast imaging within CMR can estimate peak LVOT gradients, echocardiography remains the preferred modality for this specific hemodynamic measurement. Similarly, CMR excels at clarifying the mechanism of MR, differentiating SAM-related MR from primary MR. Its volumetric assessment enables accurate and reproducible quantification of MR, particularly for the eccentric jets often seen in HCM [1,17,29,30,33].

CMR identifies and quantifies myocardial replacement fibrosis via LGE imaging. Replacement fibrosis, appearing as patchy, mid-wall, high-intensity signals on LGE within hypertrophied areas, is a critical SCD predictor (Figure 7) [34,35]. This is quantified using a > 6 SD threshold above the normal myocardium signal intensity. Although a burden of replacement fibrosis by LGE > 15% suggests increased SCD risk and potential ICD consideration, the risk associated with LGE is a continuum [36]. In contrast, interstitial fibrosis, seen as diffuse T1 elevation or faint lattice LGE, is associated with worse outcomes, including heart failure events, but has no definitive link to SCD (Figure 7) [37]. Native myocardial T1 and LGE entropies, reflecting myocardial tissue and fibrosis heterogeneity, are new and promising predictors of SCD in HCM [38,39]. Finally, abnormal myocardial perfusion due to microvascular dysfunction by CMR is an early disease marker, potentially preceding fibrosis, linked with increased non-sustained ventricular tachycardia [40,41].

### 3.4. Cardiac Computed Tomography

Cardiac computed tomography (CCT) is the preferred non-invasive method for excluding epicardial coronary artery disease and evaluating myocardial bridging (Table 2 and Table 3) [42,43]. This modality can assess LVH and even LVEF when echocardiography is suboptimal and CMR is contraindicated. Furthermore, CCT can identify anatomical abnormalities of the sub-valvular apparatus, such as apically displaced or aberrant papillary muscles (Figure 8). Emerging data suggests late iodine enhancement (LIE) on CCT could assess replacement myocardial fibrosis [44]. While still under investigation, LIE may offer an alternative for fibrosis assessment in patients unable to undergo CMR. In the preprocedural setting, CCT can help visualize septal anatomy to identify target vessels for alcohol septal ablation and to provide safety considerations for surgical myomectomy [45,46]. The drawbacks of CCT include radiation exposure, use of iodine contrast, and inferior temporal resolution.

### 3.5. Nuclear Myocardial Perfusion Imaging

Nuclear myocardial perfusion imaging (MPI), particularly positron emission tomography (PET), is valuable in HCM for assessing microvascular dysfunction once epicardial coronary artery disease is excluded (Table 2 and Table 3). PET is preferred due to its superior temporal and spatial resolution and ability to quantify myocardial blood flow [9,47]. Furthermore, asymmetric resting tracer uptake, often more intense in hypertrophied areas, can be seen in HCM patients during MPI and should prompt consideration for further evaluation if the disease is not known (Figure 9) [1,48]. Nuclear MPI can assess LVEF and diastolic function parameters such as PFR and TPFR derived from the time–activity curve. However, this modality is not routinely indicated for this purpose.

## 4. Imaging-Guided Management of HCM

### 4.1. Diagnosis and Classification

The accurate diagnosis of HCM relies heavily on a multimodality imaging approach, especially in differentiating HCM from phenocopies. Relying solely on echocardiography can lead to misdiagnosis, as these mimickers can present with similar features. Misdiagnosis can result in inappropriate treatment and potentially detrimental interventions. Similarly, determining the correct HCM phenotype, obstructive or non-obstructive, is crucial for appropriate management. Therefore, most patients with suspected HCM should undergo a comprehensive evaluation that includes TTE with complementary CMR and stress echocardiography to ensure accurate diagnosis and classification (Figure 10).

As HCM awareness continues to increase, there will be an increase in the detection of diseases before clinical symptoms, which often present with subtler changes on echocardiography requiring corroboration with other modalities. Early and accurate diagnosis, facilitated by this comprehensive approach, enables timely intervention, optimizes symptom management, allows for a broader range of treatment options, and improves risk stratification, ultimately reducing morbidity and mortality.

### 4.2. Symptom Management and Treatment of LVOTO

Patients with HCM often present with a variety of symptoms, including exertional dyspnea, fatigue, syncope, and chest pain. Diagnosing the root cause of these symptoms in HCM is particularly challenging due to the multiple potential contributing mechanisms, including systolic or diastolic dysfunction, LVOTO, MR, and ischemia from epicardial or microvascular disease. Crucially, these diverse mechanisms can present with similar symptoms, masking the underlying pathology. A multimodality imaging approach tailored to patients’ presentations can disentangle these overlapping presentations and pinpoint symptoms’ dominant contributing mechanism(s). As each of these etiologies requires a distinct management strategy, comprehensive imaging is not just about diagnosing but precisely defining the underlying pathophysiology to guide appropriate and effective treatment.

The presence of symptomatic LVOTO refractory to non-pharmacological strategies (e.g., avoidance of diuretic or pure vasodilators) informs the need to initiate beta-blockers or calcium channel blockers as an alternative [1,2,49]. As symptoms often manifest with LVOT gradients ≥ 50 mmHg at rest, during Valsalva, or with exercise, this threshold is generally used to guide treatment decisions. If symptomatic LVOTO persists despite initial therapy, escalation to CMI, disopyramide, or septal reduction therapies is recommended [50].

Comprehensive cardiac imaging is crucial when evaluating septal reduction therapies and mitral interventions for patients with HCM [1]. Simply identifying LVOTO or MR is insufficient; a thorough understanding of the underlying mechanisms is paramount. TTE and, when indicated, complementary TEE and CMR facilitate the identification of the complex interplay between anatomical and functional factors contributing to obstruction and regurgitation. Furthermore, when alcohol septal ablation is being considered, echocardiography with contrast can identify favorable coronary arterial anatomy supplying the septum [51]. By precisely characterizing these elements, imaging guides the heart team in tailoring the intervention to the patient’s needs. When surgery is considered, this may involve a combination of septal myectomy, mitral valve intervention, and papillary muscle reorientation to alleviate LVOTO and resolve MR.

### 4.3. Risk Stratification for SCD and ICD Implantation

Patients with HCM have an estimated SCD risk of approximately 0.5% per year, with younger individuals facing a higher risk than older patients [52,53,54]. Per current guidelines, all patients should undergo SCD risk stratification at initial diagnosis and every 1–2 years thereafter [2]. This involves evaluating established major clinical and imaging risk factors (Table 5). Multimodality imaging with echocardiography and CMR contributes to three major risk factors for primary prevention ICD consideration. These include a maximal wall thickness ≥ 28–30 mm, LVEF < 50%, and the presence of apical aneurysm [2]. Furthermore, CMR may guide ICD implantation when major risk factors are absent or indications for ICD remain uncertain. Specifically, extensive myocardial replacement fibrosis, quantified as ≥15% of myocardial mass on LGE imaging, may be considered (Class IIb) as an indication for primary prevention ICD implantation [36,55]. Finally, echocardiography contributes to estimating 5-year SCD risk calculation by providing measurements of left atrial diameter and peak LVOT gradient [19,56,57,58]. This risk estimation facilitates shared decision-making in primary prevention ICD implantation discussions.

Novel imaging parameters, including GLS, PET-assessed flow heterogeneity, and LGE entropy, demonstrate potential for refining SCD risk stratification [39,58,59,60,61,62]. However, further validation is required before their integration into clinical practice.

### 4.4. Assess Response to Therapy and Reverse Remodeling

TTE and stress echocardiography, when indicated, are essential for monitoring treatment response, particularly when evaluating LVOTO. Furthermore, echocardiography plays a crucial role in monitoring LVEF during CMI therapy. For example, when initiating or adjusting the dose of mavacamten, monthly TTE is recommended for the first three months (initiation phase), followed by quarterly assessments during maintenance [63]. A reduction in LVEF to <50% necessitates either dose reduction or discontinuation of the CMI [2,64]. Beyond LVEF and LVOT gradients, additional echocardiographic parameters such as LV mass index, E/e’ ratio, LAVi, and GLS can be utilized to assess for reverse remodeling during CMI use [64,65]. Although no evidence exists, the expectation is that improvements in these parameters will correlate with improved clinical outcomes. The utility of serial CMR in evaluating the effects of CMI on myocardial mass, myocardial native T1 time, and myocardial fibrosis remains unestablished.

## 5. Conclusions

Multimodality imaging is a cornerstone in the diagnosis and management of HCM. It allows for an earlier, more accurate diagnosis by having corroborating data, which also helps distinguish between morphological mimickers. A comprehensive imaging approach guides personalized treatment strategies by improving risk stratification and understanding underlying mechanisms contributing to patient symptoms. As novel imaging targets are identified and studied, more opportunities for advancing HCM care will be available.

## Figures and Tables

**Figure 1 jcm-14-02606-f001:**
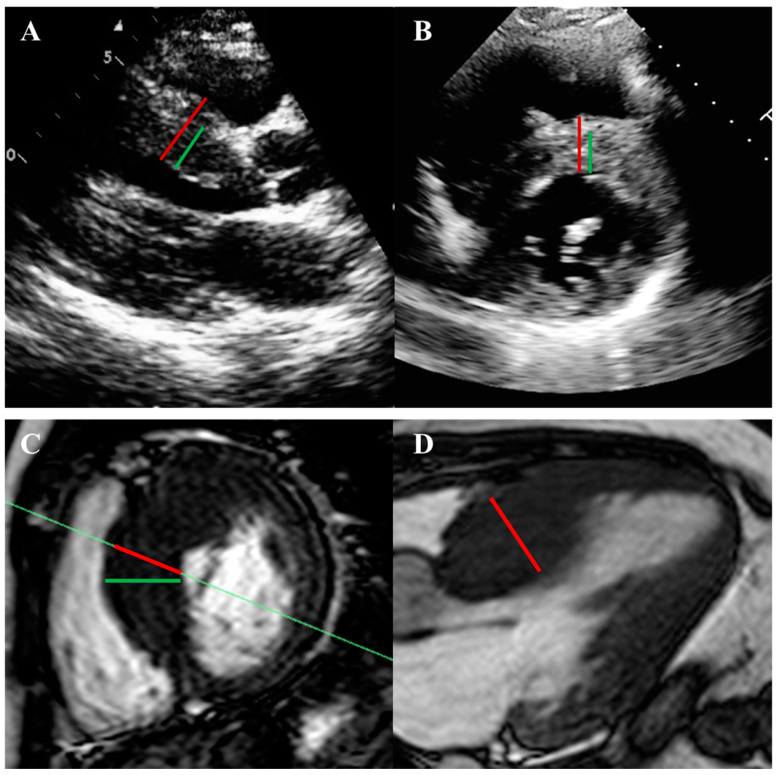
Pitfalls in the measurement of left ventricular wall thickness in echocardiography and cardiac magnetic resonance (CMR) imaging in hypertrophic cardiomyopathy (HCM). (**A**,**B**) Left ventricular septal wall thickness measurements by echocardiography in the parasternal long-axis and short-axis views, respectively. The red line (28 mm) demonstrates incorrect measurement, resulting in overestimation due to the inclusion of right ventricular trabeculae compared to the correct measurement of the green line (17 mm). (**C**,**D**) A tangential cut red line (29 mm) on CMR that leads to an overestimation of the wall thickness when measured on a 3-chamber view, compared to the correct measurement green line (24 mm).

**Figure 2 jcm-14-02606-f002:**
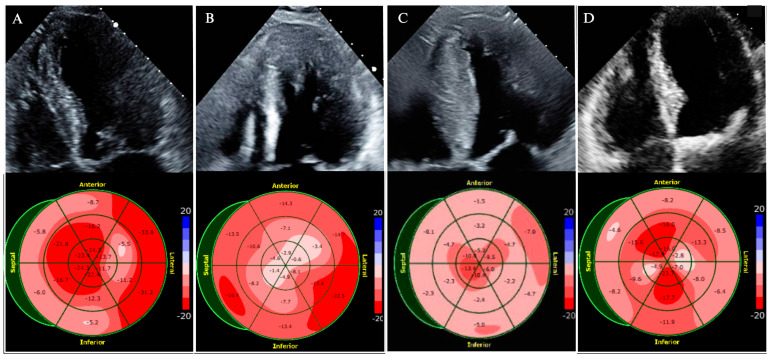
Global longitudinal strain patterns for various HCM subtypes and common mimickers. (**A**) Asymmetric septal HCM with impaired strain along the septum. (**B**) Apical HCM with impaired strain in the apex. (**C**) Cardiac amyloidosis with impaired basal-mid strain and relative apical sparing (cherry-on-top pattern). (**D**) Severe aortic stenosis with diffuse impaired longitudinal strain.

**Figure 3 jcm-14-02606-f003:**
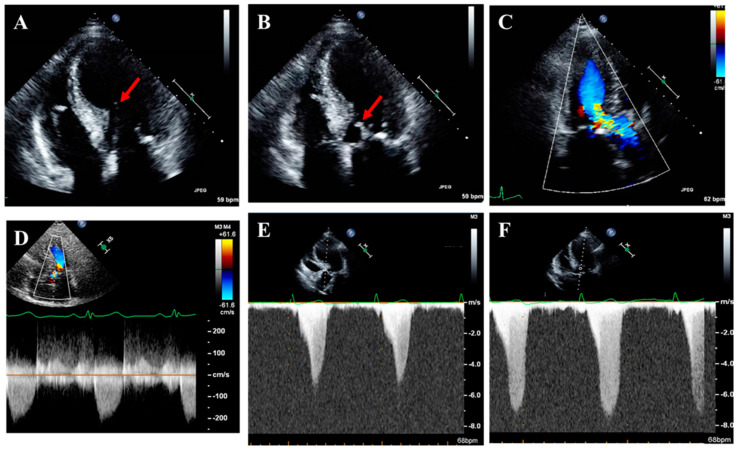
Representative echocardiography images in HCM for the comprehensive assessment of left ventricular outflow tract obstruction (LVOTO). (**A**) Depiction of an elongated anterior mitral valve (MV) leaflet (red arrow) and septal thickness as a mechanism for systolic anterior motion (SAM) of the MV. (**B**) Depiction of SAM of the MV (red arrow). (**C**) Color Doppler depicting flow acceleration in the left ventricular outflow tract (LVOT) at the level of SAM. (**D**) Continuous-wave (CW) Doppler of LVOT at rest with a gradient of 16 mmHg. (**E**) CW Doppler of LVOT with exercise showing dynamic LVOTO with a gradient of around 100 mmHg with the classic late-peaking, dagger shape. (**F**) CW Doppler with exercise and fanning of the transducer posteriorly depicting mitral signal contamination. Note the higher peak velocity, longer duration, and difference in shape compared to the LVOT signal.

**Figure 4 jcm-14-02606-f004:**
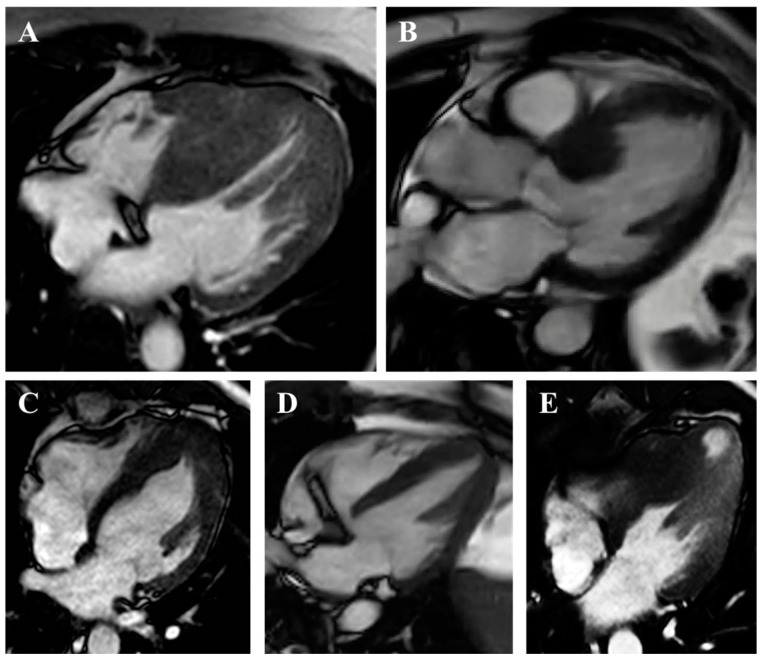
HCM phenotypes by CMR. (**A**) Reverse septum. (**B**) Sigmoid septum. (**C**) Typical apical. (**D**) Relative apical. (**E**) Mid-ventricular with apical aneurysm.

**Figure 5 jcm-14-02606-f005:**
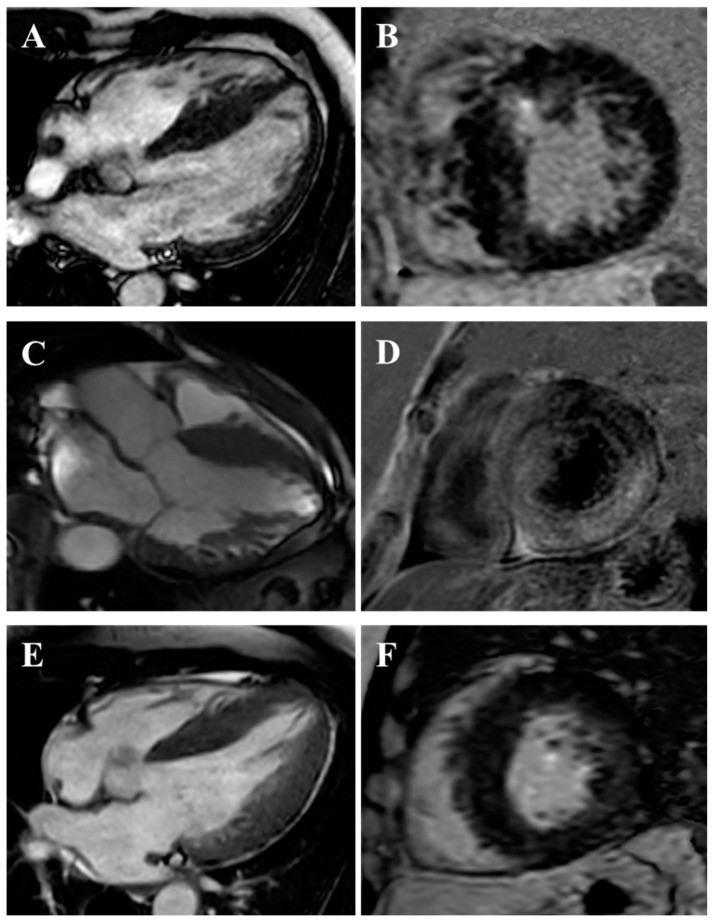
Mimickers of HCM by CMR. (**A**,**B**) PRKAG2 cardiomyopathy masquerading as HCM with asymmetric septal hypertrophy and mid-myocardial late gadolinium enhancement (LGE). (**C**,**D**) Amyloid cardiomyopathy with concentric left ventricular hypertrophy (LVH) of septal predominance with typical diffuse sub-endocardial to transmural LGE. (**E**,**F**) Hypertensive heart disease with concentric LVH with basal septal predominance without LGE.

**Figure 6 jcm-14-02606-f006:**
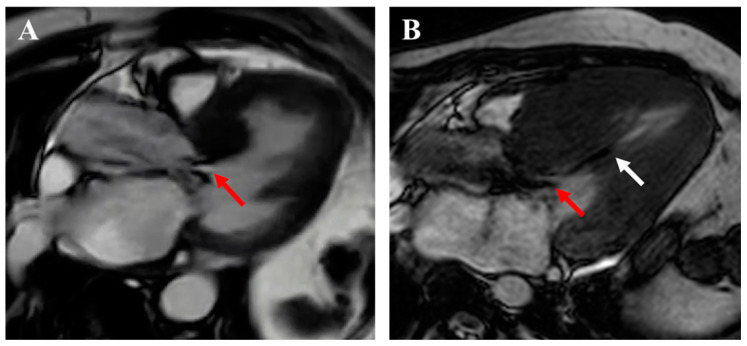
Characterization of the mechanism of obstruction in HCM by CMR. (**A**) Pure LVOTO due to SAM of the MV (red arrow). (**B**) Multi-level obstruction with LVOTO (red arrow) with SAM and intracavitary gradient (white arrow) due to LVH, prominent papillary muscles, and hyperdynamic left ventricular ejection fraction (LVEF).

**Figure 7 jcm-14-02606-f007:**
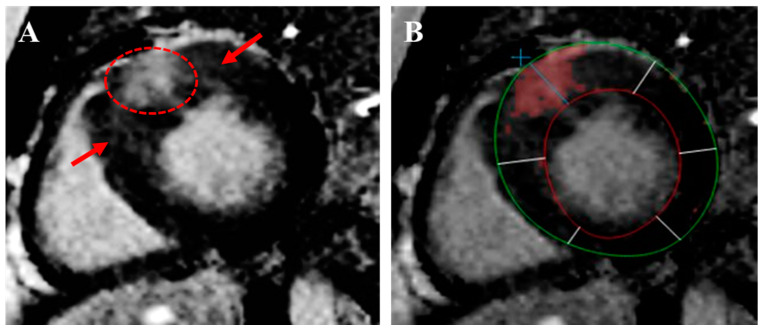
Quantification of replacement fibrosis by CMR with LGE imaging of a patient with HCM. (**A**) The dashed red circle depicts replacement fibrosis as patchy, mid-wall, high-intensity signals, while the red arrows point towards interstitial fibrosis areas appearing as faint lattice signals. (**B**) The shaded red area depicts the 6-standard-deviation method used to quantify the amount of replacement fibrosis, noting that this should not include the areas of interstitial fibrosis.

**Figure 8 jcm-14-02606-f008:**
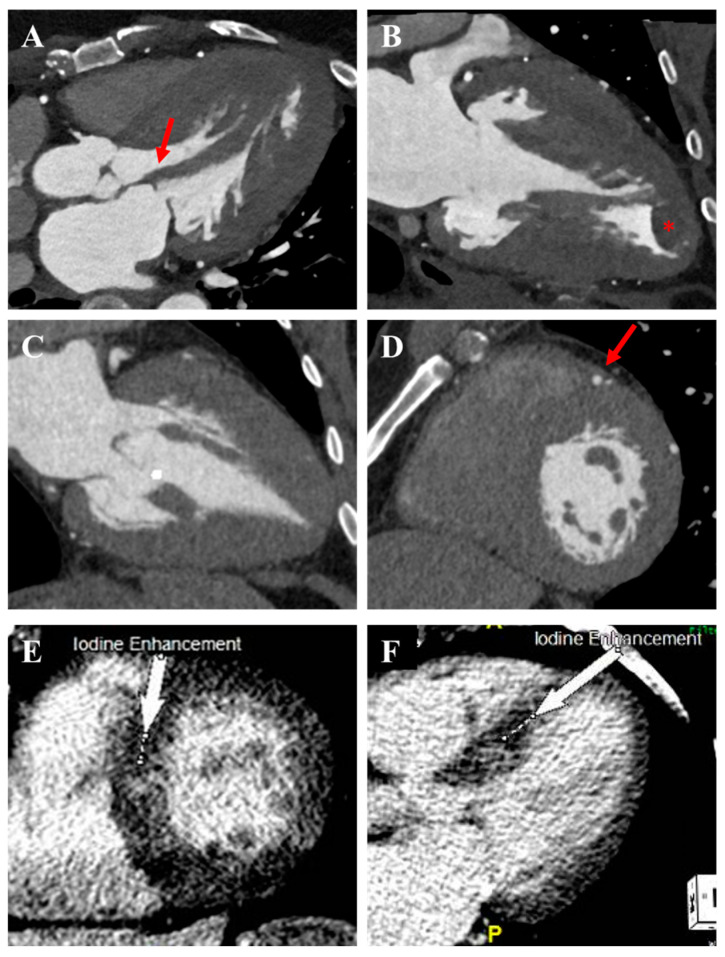
Cardiac computed tomography findings in HCM. (**A**) Asymmetric septal HCM with aberrant papillary muscle (red arrow) attached to the mitral valve contributing to LVOTO. (**B**) Mid-ventricular HCM with apical aneurysm and thrombus (*). (**C**) Apical HCM. (**D**) Myocardial bridge involving the left anterior descending artery (red arrow). (**E**,**F**) Asymmetric septal HCM with mid-myocardial late iodine enhancement (white arrows).

**Figure 9 jcm-14-02606-f009:**
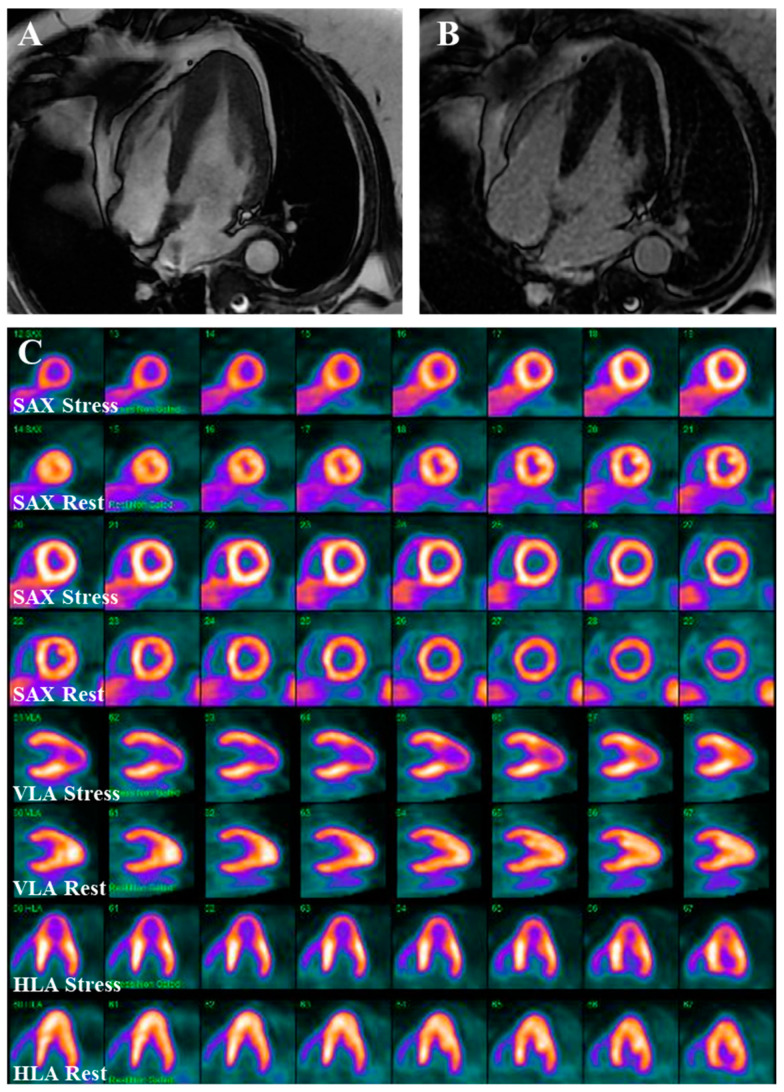
Nuclear myocardial perfusion imaging (MPI) with positron emission tomography (PET) in apical HCM. (**A**,**B**) CMR images of a patient with apical HCM without epicardial coronary artery disease (not demonstrated in these images). (**C**) The corresponding MPI with PET depicting prominent resting tracer uptake at the apex consistent with the diagnosis of apical HCM with stress-induced apical ischemia due to microvascular disease. SAX: short axis; VLA: vertical long axis; HLA: horizontal long axis.

**Figure 10 jcm-14-02606-f010:**
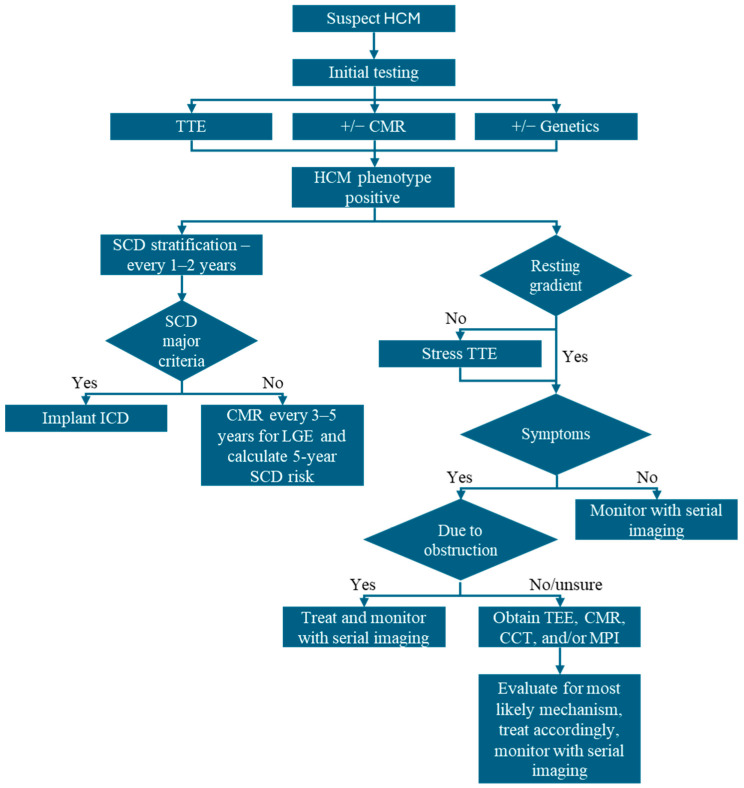
Imaging algorithm for patients with suspected and confirmed HCM. CCT: cardiac computed tomography; CMR: cardiac magnetic resonance; HCM: hypertrophic cardiomyopathy; ICD: implantable cardioverter defibrillator; MPI: myocardial perfusion imaging; TEE transesophageal echocardiography; TTE: transthoracic echocardiography; SCD: sudden cardiac death.

**Table 1 jcm-14-02606-t001:** Diagnostic criteria for HCM in adults.

Guideline-Supported Criteria	Supporting Findings
LV end-diastolic wall thickness ≥ 15 mm in any segment not entirely explained by another condition	LVH that predominantly involves the basal septum or apex, sparing the posterior wall SAM of MV and LVOTO Apically displaced papillary muscle Elongated anterior mitral valve leaflet
LV end-diastolic wall thickness 13–14 mm with a positive pathogenic genetic variant or in primary relatives of patients with HCM	
In children: Indexed LV end-diastolic wall thickness z score ≥ 2	
**Emerging criteria**	
Demographically adjusted LVH	
Indexed apical wall thickness > 5.6 mm/m^2^ in apical HCM	

HCM: hypertrophic cardiomyopathy; LV: left ventricular; LVH: left ventricular hypertrophy; LVOTO: left ventricular outflow tract obstruction; MV: mitral valve; SAM: systolic anterior motion.

**Table 2 jcm-14-02606-t002:** Indication and frequency of imaging multimodality in HCM.

Modality	When to Obtain?	When to Repeat?
TTE	Initial evaluation Surveillance on cardiac myosin inhibitors	1–2 years, or functional decline As per cardiac myosin inhibitor protocol
TEE	Uncertain LVOTO mechanism despite TTE Uncertain MR mechanism/severity despite TTE Procedural planning and intraprocedural	Personalized
Stress TTE	Initial evaluation if no LVOTO (<30 mmHg) at rest or with Valsalva Symptomatic patients with LVOTO < 50 mmHg at rest or with Valsalva	2–3 years, or functional decline
CMR	Initial evaluation when TTE is non-diagnostic/equivocal or when differentiation from mimickers is needed Uncertain LVOTO mechanism despite TTE Uncertain MR mechanism/severity despite TTE SCD risk remains uncertain Procedural planning	3–5 years for SCD risk if no ICD present
CCT	Evaluation of obstructive epicardial CAD Evaluation of LVH when TTE is non-diagnostic and CMR not available/contraindicated Assessment of replacement fibrosis (late iodine enhancement) when CMR is not available/contraindicated Procedural planning	Personalized
Nuclear MPI	Evaluation of microvascular ischemia after excluding epicardial CAD	Personalized

CAD: coronary artery disease; CCT: cardiac computed tomography; CMR: cardiac magnetic resonance; ICD: implantable cardioverter defibrillator; LVH: left ventricular hypertrophy; LVOTO: left ventricular outflow tract obstruction; MPI: myocardial perfusion imaging; MR: mitral regurgitation; TEE: transesophageal echocardiography; TTE: transthoracic echocardiography; SCD: sudden cardiac death.

**Table 3 jcm-14-02606-t003:** Imaging modalities and their capability to evaluate imaging targets *.

Modality	Imaging Targets								
	LVH	Systolic Function	Diastolic Function	Apical Aneurysm	LVOTO (Gradient)	MR/LVOTO Mechanisms	Epicardial CAD	MVD	Fibrosis
TTE	+	+	++	+/++ with UEA	++	+	−	−	−
TEE	++	+	+	+	++	+++	−	−	−
Stress TTE	+	+	+++ (exercise diastology)	+/++ with UEA	+++ (latent LVOTO)	++ (dynamic MR with exercise)	+	+ (LAD CFR)	−
CMR	+++	+++	+	+++	+	+++	++ (Stress CMR)	++	++ (LGE)
CCT	+++	++	−	+++	−	+/− (anatomical abnormalities)	+++	+/− (CCT perfusion)	+ (LIE)
Nuclear MPI	+/− (Asymmetric uptake)	+	+/− (PFR/TPFR)	−	−	−	++	+++	−

CAD: coronary artery disease; CCT: cardiac computed tomography; CFR: coronary flow reserve; CMR: cardiac magnetic resonance; LAD: left anterior descending artery; LIE: late iodine enhancement; LGE: late gadolinium enhancement; LVH: left ventricular hypertrophy; LVOTO: left ventricular outflow tract obstruction; MPI: myocardial perfusion imaging; MR: mitral regurgitation; MVD: microvascular disease; PFR: peak filling rate; quant: quantification; TEE transesophageal echocardiography; TPFR: time to peak filling rate; TTE: transthoracic echocardiography; UEA: ultrasound enhancing agent. * The number of + indicates that the imaging modality is superior in evaluating the target compared to other imaging modalities. Even if an imaging modality demonstrates capability or superiority (+++) for a particular target, it may not be the preferred initial choice or routinely recommended. Table 2 provides guidance on appropriate indications and repeat imaging frequency.

**Table 4 jcm-14-02606-t004:** Instructions on discrimination of mitral regurgitation from left ventricular outflow tract signal on echocardiography.

Steps	Practical Guidance
1. Identify MR signal with CW Doppler	Begin by placing the CW Doppler beam through the easily identifiable MR jet on color Doppler. Obtain and save the signal.
2. Maneuver toward the LVOT	With the CW Doppler, slowly sweep the transducer anteriorly toward the LVOT.
3. Observe for LVOT signal	Look for a signal that: ● Starts later in systole ● Has a late-peaking, dagger-shaped appearance ● Has a shorter duration than the MR signal ● Has a lower peak velocity than the MR signal
4. PW Doppler mapping	Use PW Doppler to map velocities around the LVOT, starting deep in the LV cavity and moving toward the aorta. A sudden increase in the velocity helps confirm the LVOT signal’s location.
5. Compare and contrast	Compare the saved MR signal with the suspected LVOT signal. Note the differences in timing, shape, velocity, and duration.
6. Valsalva	If uncertain, perform a Valsalva maneuver while monitoring the LVOT signal. An increase in velocity and a more pronounced dagger shape support the presence of a dynamic LVOTO.

CW: continuous wave; LVOT: left ventricular outflow tract; LVOTO: left ventricular outflow tract obstruction; MR: mitral regurgitation; PW: pulse wave.

**Table 5 jcm-14-02606-t005:** Specific imaging parameters for sudden cardiac death risk assessment.

Parameter	Cut-Off
**Major**	
LV wall thickness	≥28–30 mm
Apical aneurysm	Presence
LVEF	<50%
**Non-major**	
LGE	≥15%
**5-year risk estimation**	
LVOT gradient	Continuous, but higher risk at >30 mmHg
Left atrial diameter	Continuous
**Investigational**	
Left atrial volume index	>34 mL/m^2^
GLS	>−15%
Flow heterogeneity	≥1.85
LGE entropy	≥5.873

GLS: global longitudinal strain; LGE: late gadolinium enhancement; LV: left ventricular; LVEF: LV ejection fraction; LVOT: left ventricular outflow tract.

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
