# Peer review of "A Practical Approach to Multimodality Imaging in Hypertrophic Cardiomyopathy"

_jcm, 2025, doi:10.3390/jcm14082606_

Round 1
Reviewer 1 Report
Comments and Suggestions for Authors
This is a well-written and thoroughly researched review that provides a comprehensive overview of this clinically relevant topic.
-
My main suggestion concerns the overall structure of the manuscript, which could benefit from better alignment with the clinical presentation and diagnostic workflow of patients. This adjustment would improve the practical use for clinicians.
Consider using the following framework for the major headlines, (e.g.):-
Establishing the Diagnosis
-
Pitfalls and Challenges
-
Risk Stratification
-
Epicardial CAD / MVD
-
Imaging-Guided Management of HCM
-
-
Consider adding a brief summary at the end of each major section
-
Consider to include a paragraph discussing the (possible lack of) availability of advanced imaging modalities (e.g., CMR and CT) across different healthcare settings
-
In Figure 1, correct the red measurement line in Panel C
- consider deleting `common` Line 23
- line 453 ->` LVEV`, please correct
Author Response
Comment 1: My main suggestion concerns the overall structure of the manuscript, which could benefit from better alignment with the clinical presentation and diagnostic workflow of patients. This adjustment would improve the practical use for clinicians.
Response 1: We thank the reviewer for considering the structure of our manuscript and advice on improvement. However, the current structure was chosen to provide an imaging focused assessment of HCM. There are other reviews that focus on the clinical presentation and the general diagnostic workflow of HCM, however the focus of our manuscript is on the use of imaging in HCM. We believe that the current structure enhances practical use by briefly highlighting the imaging targets, then providing a detailed assessment by imaging modality, and lastly discussing the image-guided management of HCM.
Comment 2: Consider adding a brief summary at the end of each major section.
Response 2: We thank the reviewer for this suggestion. However, the sections of our manuscript are manageable and we believe that adding a summary at the end of each major section will increase redundancy without improving readability. Additionally, our tables are thoughtfully created to provide a summary of the sections in a concise manner.
Comment 3: Consider to include a paragraph discussing the (possible lack of) availability of advanced imaging modalities (e.g., CMR and CT) across different healthcare settings.
Response 3: We thank the reviewer for this insightful comment. Given this review focuses on the technical aspects of multimodality imaging in HCM we don’t believe we can adequately discuss the availability of the modalities across different healthcare settings. The diagnosis and management of HCM mostly occurs in larger medical centers which should have access to CMR and CCT. In the future, we will consider studying the healthcare disparities in HCM care for which we can delve into the availability of advanced imaging in different settings.
Comment 4: In Figure 1, correct the red measurement line in Panel C.
Response 4: We thank the reviewer for this suggestion. We also considered this prior to submission as the red line in Figure 1 panel C is confusing, however there is an underlying thin green line that is part of the CMR image which we cannot remove. Regardless, we have fixed the red line to only include the wall.
Comment 5: consider deleting `common` Line 23.
Response 5: We agree and have deleted the word “common.”
Comment 6: line 453 ->` LVEV`, please correct.
Response 6: This has been corrected.
Reviewer 2 Report
Comments and Suggestions for Authors
Thank you very much for this interesting manuscript
I have some suggestions
Title: add Review Article over the title
In the abstract and at the end of the introduction: add a sentence that clearly states that this is a narrative review.
line 121: "Although common in HCM, SAM can occur in other conditions" add some examples, "in other conditions, such as...."
line 268: "Pitfalls such as foreshortening and misinterpretation of normal variants must be avoided"; please add some examples about normal variants.
If you have 4D flow images, please addthem in the manuscript.
If you have T1mapping images, please add them in the manuscript.
Line 344: please check if the acronym NSVT was written before in the text.
3.4. Cardiac computed tomography secrion: I ask you to add at least a sentence about myocardial extracellular volume measurement using cardiac computed tomography. You could add the reference written below or another one. Muthalaly RG, Abrahams T, Lin A, Patel K, Tan S, Dey D, Han D, Tamarappoo BK, Nicholls SJ, Nerlekar N. Myocardial extracellular volume measurement using cardiac computed tomography. Int J Cardiovasc Imaging. 2024 Nov;40(11):2237-2245. doi: 10.1007/s10554-024-03226-4.
Many thanks
Author Response
Comment 1: Title: add Review Article over the title
Response 1: We will leave it to the journal editor team to determine how the article is labelled per their journal standards.
Comment 2: In the abstract and at the end of the introduction: add a sentence that clearly states that this is a narrative review.
Response 2: We thank the reviewer for this suggestion, however the last sentence in both the abstract and the introduction already states that this is a review.
Comment 3: line 121: "Although common in HCM, SAM can occur in other conditions" add some examples, "in other conditions, such as. "
Response 3: We thank the reviewer for this comment, we have added examples of other conditions that can cause systolic anterior motion of the mitral valve. The changes are highlighted.
Comment 4: line 268: "Pitfalls such as foreshortening and misinterpretation of normal variants must be avoided"; please add some examples about normal variants.
Response 4: We thank the reviewer for this suggestion, we have added an example with the changes highlighted.
Comment 5: If you have 4D flow images, please add them in the manuscript. If you have T1 mapping images, please add them in the manuscript.
Response 5: We thank the reviewer for this comment. However, we do not have 4D flow images and we believe the current CMR images highlight the important aspects of HCM.
Comment 6: Line 344: please check if the acronym NSVT was written before in the text.
Response 6: We thank the reviewer for pointing this out, the acronym has been replaced by the full phrase.
Comment 7: Cardiac computed tomography section: I ask you to add at least a sentence about myocardial extracellular volume measurement using cardiac computed tomography. You could add the reference written below or another one.
Response 7: We thank the reviewer for this comment. While extracellular volume (ECV) measurement using CCT is exciting, we did not mention it in our review as ECV is used to quantify interstitial fibrosis which does not have a definitive link to sudden cardiac death in patients with HCM as compared to replacement fibrosis which does. Thus, for increased clarity we have focused on techniques used to quantify replacement fibrosis.